# Evaluation of Antimicrobial Effect of Air-Polishing Treatments and Their Influence on Human Dental Pulp Stem Cells Seeded on Titanium Disks

**DOI:** 10.3390/ijms22020865

**Published:** 2021-01-16

**Authors:** Rosanna Di Tinco, Giulia Bertani, Alessandra Pisciotta, Laura Bertoni, Jessika Bertacchini, Bruna Colombari, Enrico Conserva, Elisabetta Blasi, Ugo Consolo, Gianluca Carnevale

**Affiliations:** 1Department of Surgery, Medicine, Dentistry and Morphological Sciences with Interest in Transplant, Oncology and Regenerative Medicine, University of Modena and Reggio Emilia, 41125 Modena, Italy; rosanna.ditinco@unimore.it (R.D.T.); giulia.bertani@unimore.it (G.B.); alessandra.pisciotta@unimore.it (A.P.); laura.bertoni@unimore.it (L.B.); jessika.bertacchini@unimore.it (J.B.); bruna.colombari@unimore.it (B.C.); enrico.conserva@unimore.it (E.C.); elisabetta.blasi@unimore.it (E.B.); ugo.consolo@unimore.it (U.C.); 2Operative Unit of Dentistry and Maxillofacial Surgery, Department Integrated Activity-Specialist Surgeries, University-Hospital of Modena, 41125 Modena, Italy

**Keywords:** glycine, tagatose, air-polishing powders, anti-microbial effect, titanium implants, hDPSCs, stem cell properties

## Abstract

Dental implants are one of the most frequently used treatment options for tooth replacement, and titanium is the metal of choice due to its demonstrated superiority in resisting corrosion, lack of allergic reactions and mechanical strength. Surface roughness of titanium implants favors the osseointegration process; nevertheless, its topography may provide a suitable substrate for bacterial biofilm deposition, causing peri-implantitis and leading to implant failure. Subgingival prophylaxis treatments with cleansing powders aimed to remove the bacterial accumulation are under investigation. Two different air-polishing powders—glycine and tagatose—were assayed for their cleaning and antimicrobial potential against a *Pseudomonas* biofilm and for their effects on human dental pulp stem cells (hDPSCs), seeded on sandblasted titanium disks. Immunofluorescence analyses were carried out to evaluate cell adhesion, proliferation, stemness and osteogenic differentiation. The results demonstrate that both the powders have a great in vitro cleaning potential in the early period and do not show any negative effects during hDPSCs osteogenic differentiation process, suggesting their suitability for enhancing the biocompatibility of titanium implants. Our data suggest that the evaluated cleansing systems reduce microbial contamination and allow us to propose tagatose as an adequate alternative to the gold standard glycine for the air-polishing prophylaxis treatment.

## 1. Introduction

Dental implant procedures are considered as a common treatment that allows the replacement of absent or lost teeth. Among the different materials that might be used for the manufacture of oral implants, titanium represents the most common dental implant material clinically used thanks to its properties: good strength, elasticity similar to the jawbone and surface morphology favoring the process of osseointegration [1,2]. Indeed, osseointegration might be influenced by geometry, micro- and nanotopography characteristics and chemical composition of the surfaces, as well as by the surface treatments [3,4,5]. In this regard, it is well known that biomimetic surfaces are able to promote cellular adhesion and to accelerate the osteogenic commitment of resident osteoprogenitor cells/stem cells, resulting in the promotion of the early osseointegration [6]. However, despite titanium implants having high rates of long-term survival (>10 years), biological complications associated with dental implants could occur in the peri-implant area, causing the failure of treatment [7].

These clinical complications, named peri-implant mucositis and peri-implantitis, mainly refer to inflammatory conditions that are induced by the accumulation of bacterial biofilm around the implant. Such a pathological condition occurring in tissues around implant sites leads to inflammation of the connective tissue and, gradually, to a progressive loss of the supporting bone [8,9,10]. Moreover, it is well-known that peri-implantitis has an infectious etiology through the development of biofilm composed of a plethora of well-known oral pathogens, including *Porphyromonas gingivalis* and *Tannerella forsythia*, as well as other opportunistic pathogens such as *Pseudomonas aeruginosa* and *Staphylococcus aureus* [11].

Therefore, in order to optimally preserve the success of the implant after a bacterial contamination, it is crucial to periodically cleanse dental implants using prophylaxis treatments [12,13,14,15]. In this regard, the use of specific powders facilitates the removal of both supragingival and subgingival plaque, thus hopefully preventing the occurrence of peri-implantitis and its complications, including implant failure. To date, several studies have proven the efficacy of different powders for supragingival prophylaxis treatment, such as sodium bicarbonate and calcium carbonate, besides the gold standard glycine [16,17,18,19,20].

At the same time, glycine is considered the gold standard also in the subgingival prophylaxis treatment [21,22,23], although studies on further powders are lacking. For these reasons, recent studies have been focused on new compounds characterized by a cleaning efficacy higher than or at least similar to that of glycine. Importantly, optimal air-polishing compounds are expected to be biocompatible [24,25,26] in order to ensure the maintenance of biological properties of resident osteoprogenitor cells/ecto-mesenchymal stem cells involved in histointegration processes. In this regard, the aim of the present study was to investigate a novel experimental powder tailored for the development of subgingival prophylaxis treatments, in terms of effects on the biological properties of human dental pulp stem cells (hDPSCs). These stem cells have a peculiar embryological origin from the neural crest, which is common to bone progenitor cells involved in histointegration processes [27]. In light of these properties, hDPSCs may provide a suitable cell source to assess the cell adhesion, proliferation and osteogenic differentiation when seeded on biomimetic/nanostructured titanium disks previously cleansed with different powders.

## 2. Results

### 2.1. Effects of the Decontamination Procedure on the Formation and Persistence of Microbial Biofilm onto Titanium Disks

As depicted in Table 1, our data reveal the presence of a consistent signal, indicating that an early biofilm is indeed produced on the titanium disks. Following the cleaning treatments, the residual signal detected on the experimental groups was significantly reduced compared to the control group (* *p* < 0.05, GLY vs. CTRL and TAGAT vs. CTRL). In particular, a more than 6-fold and 4-fold decrease was observed following glycine (GLY) or tagatose (TAGAT) treatment, respectively. These data indicate that both treatments have a good cleansing efficacy against *P. aeruginosa*-contaminated titanium disks. Furthermore, when after 30 h of incubation the persistent biofilm was assessed, no differences were observed between tagatose and the control group, while a 1.7-fold decrease was detected when comparing the glycine-treated with the control group. Finally, when comparing the persistent biofilm with the early biofilm, we found that the glycine treatment allowed biofilm increase of 18-fold, while the control group and the tagatose-treated group increased approximately 33-fold. Overall, these data indicate that the glycine and tagatose cleansing systems effectively removed *Pseudomonas* biofilm from titanium disk, yet to a different extent.

### 2.2. Immunophenotypical Characterization of hDPSCs

After isolation from human dental pulp specimens, cells underwent immune-selection against STRO-1 and c-Kit stemness markers in order to obtain a purer stem cell population of hDPSCs. Then, flow cytometry analysis was performed on STRO-1^+^/c-Kit^+^ hDPSCs, expanded upon confluence at passage 1, to assess the expression of MSC surface markers. As reported in Figure 1, STRO-1^+^/c-Kit^+^ hDPSCs were labeled positive against CD73, CD90, CD105 and, to a lesser extent CD34, while showing no labeling against CD45 and HLA-DR, in accordance with previous reports [27].

### 2.3. Titanium Surface Characterization

In order to evaluate whether the air-polishing treatments alter the titanium surface nanotopography, SEM analysis was carried out on titanium surfaces from the three experimental groups. As shown in Figure 2, GLY and TAGAT air-polishing treatments did not induce any relevant alteration of the surface nanotopography.

### 2.4. Cell Morphology and Proliferation of hDPSCs on Titanium Disks

Immune-selected hDPSCs were seeded on titanium disks previously air-polished with two different experimental powders, i.e., GLY and TAGAT; then, cell morphology and proliferation were evaluated at 24, 48, 72 h and 7 days of culture and compared to hDPSCs seeded on untreated titanium disks (CTRL). As shown by phalloidin stain in Figure 3A, in GLY group, hDPSCs showed an elongated spindle-shaped morphology when compared to the CTRL group and to the TAGAT group, as early as after 24 h of culture. Similarly, hDPSCs from TAGAT group exhibited a more typical fibroblast-like morphology after 48 h of culture, which was maintained for the whole culture time, besides showing a homogeneous cell spreading over the surface. After 7 days of culture, hDPSCs did not show any difference in terms of cell morphology and distribution, suggesting that the two different air-polishing powders did not alter the cell morphology and nanotopography of titanium surface (Figure 3B). Likewise, as far as cell adhesion and proliferation were concerned, no statistically significant difference was observed among the experimental groups.

At the same time, the evaluation of stemness markers was performed in order to confirm the maintenance of stem cell properties after 48 h of culture on titanium disks previously air-polished with GLY and TAGAT (Figure 4). As shown in Figure 4, the expression of STRO-1 and c-Kit markers was observed in all the experimental groups, thus demonstrating that TAGAT air-polishing treatment did not affect the biological/stemness properties of hDPSCs, similarly to GLY treatment.

### 2.5. Analysis of Angiogenic Phenotype of hDPSCs after Different Air-Polishing Treatments: VEGF Expression

It is well known that hDPSCs hold an angiogenic potential able to strongly support tissue regeneration in vivo [28,29,30], which is the consequence of hDPSCs localization in the perivascular region of human dental pulp [27,31]. Based on this evidence, the expression of VEGF was investigated in hDPSCs seeded on titanium disks, following air-polishing treatment with the two different cleansing powders. In Figure 5, confocal immunofluorescence analysis revealed that VEGF was expressed by hDPSCs after each surface polishing treatment and that, as highlighted by pseudocolor images, the air-polishing prophylaxis enhanced VEGF expression when compared to control conditions.

### 2.6. Osteogenic Differentiation

Since the success of titanium implants relies on the early osseointegration process, our study also aimed to evaluate whether the two different air-polishing powders may affect the osteogenic differentiation potential of hDPSCs cultured on titanium surfaces. After 3 weeks of induction, immunofluorescence analysis was performed to investigate the expression of bone extracellular matrix-associated markers, i.e., OCN and RUNX-2. Images in Figure 6 show that hDPSCs cultured on titanium surfaces after air-polishing treatments were able to undergo the osteogenic commitment, as demonstrated by the positive labeling against both earlier and later markers.

## 3. Discussion

Dental implants procedures are considered as a common treatment that allows the replacement of missing or hopeless teeth. The clinical success of oral implants is related to their osseointegration, a process in which osteoprogenitors/stem cells are involved [32,33]. The osseointegration process is strictly related to the characteristics of implant material, including geometry and surface topography [34,35,36]. In this regard, the sandblasting procedures relying on the use of resorbable or non-resorbable blasting media allow one to obtain different levels of surface roughness affecting the micro- and nanotopography profile of dental implants [37]. The rough surfaces have the advantage of mimicking the histomorphology of jawbone favoring cell colonization, osseointegration and, consequently, bone ongrowth [19,38,39,40], but at the same time, they favor the colonization and accumulation of a bacterial biofilm in a surface-roughness-independent manner. This plaque accumulation leads to the development of peri-mucositis up to peri-implantitis [41,42,43,44].

As reported by Schminke et al., approximately 30% of patients with dental implants develop peri-implantitis, a pathological condition occurring in tissues around dental implants, characterized by local infection and inflammation in the proximal connective tissue, which leads to a progressive loss of supporting bone [45]. Therefore, cleansing treatments able to remove the bacterial biofilm, as the initial causative agent of peri-implant diseases, are needed. Traditionally, biofilm removal has mostly relied on the use of ultrasonic/sonic devices, plastic hand instrumentation, titanium brushes, diode laser and air polishing [46]. The traditional supragingival air polishing powder used on natural dentition is sodium bicarbonate, whose limitations are represented by root structure loss, making them not the ideal cleansing candidate for patients with periodontal disease, as there is a high incidence of coexisting gingival recession in these patients. Therefore, even though benefits exist for the use of conventional sodium bicarbonate air polishing, studies have highlighted that it is not suitable for effective debridement of biofilm from root surfaces or subgingival surfaces [25,47]. Based on this evidence, investigations have been carried out to identify new cleansing powders not only for supragingival removal but also for subgingival surfaces.

In this regard, glycine powder has been identified as the main air-polishing agent. Glycine is a non-essential amino acid; its particle size is significantly smaller than sodium bicarbonate powders, and it causes decreased gingival trauma when compared to hand instrumentation and sodium bicarbonate powder, thus resulting in a low abrasive air polishing treatment free of risk to damage oral tissues, such as dentin and cementum [22,48]. For these reasons, further studies have been focused on new compounds characterized by higher or at least similar benefits compared to glycine.

It is noteworthy that the requirements of these compounds are not solely based on a good cleansing efficacy but also on their capability to neither modify the implant surface roughness nor influence the biological properties of the cells involved in the subsequent osseointegration process [19]. Several in vitro studies evaluate these aspects using cell lines, such as Sarcoma osteogenic cells (Saos-2) and MC3T3E1, which have no role in osseointegration process and do not mimic the physiological conditions in the implant site [49,50]. From the embryological point of view, the craniofacial bone has a peculiar origin from the neural crest, and therefore, it would be preferable to choose a cell source sharing the same embryological origin. For such reasons, in this study, the experiments were performed using hDPSCs immune-selected against the stemness markers c-Kit and STRO-1. hDPSCs are neural crest-derived stem cells residing in the loose connective tissue entrapped in the pulp chamber of the teeth. Thanks to their origin, hDPSCs are able to differentiate towards different lineages such as osteogenic, adipogenic and myogenic commitments [27]. Moreover, as is well-established in the literature, hDPSCs reside in close proximity to dental pulp vessels and express typical angiogenic markers, including Vascular Endothelial Growth Factor (VEGF), thus allowing them to be defined as pericytes-like cells. In particular, in vivo studies of tissue regeneration have revealed the ability of hDPSCs to support host tissue regeneration by promoting neo-angiogenesis [27,28,29,30].

Based on these premises, hDPSCs were seeded on titanium disks previously cleansed with two different powders suitable for a subgingival prophylaxis treatment. Therefore, the aim of the present study was to evaluate not only the cleansing efficacy of the two powders but also how they might influence the biological properties of hDPSCs. The two different powders used were glycine and tagatose. The latter one is a ketohexose, characterized by 92% of the sweetness but 38% of the calories of sucrose. It has been mostly investigated for its beneficial effect in reducing the risk of caries formation by inhibiting *S. mutans* biofilm formation [51].

Hence, tagatose might be an attractive and alternative cleansing powder to glycine, which has represented so far the gold standard in dentistry for both supra- and subgingival prophylaxis treatments.

By a bioluminescence-based prototype, we provided direct evidence that titanium disks contaminated by *Pseudomonas* promptly develop a microbial biofilm onto their surface, roughly mimicking what may occur in vivo. Moreover, upon treatment of the contaminated disks with GLY and TAGAT, biofilm was affected to a different extent. In particular, a good immediate cleansing efficacy occurred, with an appreciable biofilm reduction being observed by GLY and TAGAT treatments, respectively. As predictable, at later times, microbial regrowth and biofilm persistence were detected in all the groups. Although TAGAT treated disks were comparable to untreated controls, GLY treatment resulted in a still appreciable impairment of *Pseudomonas* load at time 30 h. Taken together, these findings imply that both GLY and TAGAT act as cleansing system, though to a different extent, especially in the early period. In this respect, it should be noted that the present data were obtained by employing *Pseudomonas aeruginosa*, a pathogen well-known for producing a highly structured, robust and resistant biofilm. Even though none of the two provided a permanent solution as a titanium decontaminating agent, we may envisage a different scenario in patients where even a partial reduction of microbial contamination may have a positive clinical outcome, since the host immune reaction may efficiently counteract the residual pathogen and definitively clear the local infection. In any case, it is well known that, in daily clinical practice, these prophylaxis treatments are associated with other cleansing methods, with the aim of enhancing the chance of efficiently decontaminating implants [19].

Regarding the influence of these cleansing powders on the biological properties of hDPSCs, data showed that both GLY and TAGAT did not alter the expression of the stemness markers STRO-1 and c-Kit. As previously reported, the immune selection against STRO-1 and c-Kit markers allows one to obtain a purer stem cell population from human dental pulp [51]. In particular, hDPSCs expressing both these stemness markers were demonstrated to have a high tendency to commit towards mesenchyme-derived cell types, including the osteogenic lineage [52,53]. Therefore, the air polishing prophylaxis treatments with GLY and TAGAT proved to be effective in maintaining the stemness of hDPSCs and, consequently, their differentiation potential. As a matter of fact, hDPSCs cultured on titanium disks formerly air-polished with GLY and TAGAT, respectively, were able to reach the osteogenic differentiation, as confirmed by the expression of both RUNX-2 and OCN. Interestingly, an increased expression of VEGF was detected in hDPSCs as early as after 7 days of culture on titanium disks cleansed with GLY and TAGAT, with respect to the control group. These data strengthen the evidence that RUNX-2 and VEGF are directly related to each other. In particular, in the osseointegration process, a key role is played by the formation of new blood vessels, suggesting a strict relationship between angiogenesis and osseointegration processes [54].

In light of our findings, it may be argued that tagatose might provide an alternative efficient air polishing treatment with respect to glycine. As a matter of fact, tagatose treatment does not alter the surface roughness of the implant and promotes neo-angiogenesis, which plays a key role in osteogenic differentiation of progenitor cells. Together with a good cleansing efficacy, these features allow us to propose tagatose air polishing powder as a suitable treatment since it does not alter the hDPSCs biological properties including their osteogenic potential.

Our data suggest that both glycine and tagatose act as decontaminating agents against a *Pseudomonas* biofilm preformed on the titanium disks. Furthermore, tagatose does not influence the biological properties of stem cells showing similar efficacy to the gold standard, glycine.

## 4. Materials and Methods

### 4.1. Microbial Strain

The bioluminescent *Pseudomonas aeruginosa* strain P1242 (BLI-Pseudomonas), previously engineered in order to express the luciferase gene and substrate under the control of a constitutive P1 integron promoter 2 was used [55]; such cells, only when viable, constitutively produce a detectable bioluminescent signal. Bacteria from −80 °C glycerol stocks were initially seeded onto Tryptic Soy Agar (TSA; Oxoid, Milan, Italy) plates and incubated overnight at 37 °C; then, isolated colonies were collected, transferred into 10 mL of TSB and allowed to grow overnight at 37 °C under gentle shaking. Bacterial concentration was then assessed by optical density using a McFarland standard curve and appropriately diluted.

#### 4.1.1. Microbial Growth and Early Biofilm Formation

Overnight cultures of BLI-*Pseudomonas* (diluted to 10^6^/mL) in TSB plus 2% sucrose were seeded (180 μL/well) in a 96-well black plate containing 1 disk/well; in parallel wells, BLI-*Pseudomonas* was seeded also without the titanium disks. To allow microbial growth and biofilm formation onto the disks, the plate was incubated at 35 °C for 15 h. Then, the disks were washed twice with warm phosphate-buffered saline (PBS) (EuroClone, Wetherby, UK), transferred into new wells and assessed for bioluminescence by the Fluoroskan Luminescence reader (Thermo Fisher Scientific, Waltham, MA, USA). Such values were taken as a measure of the early biofilm (15 h) formed onto titanium disk surfaces.

#### 4.1.2. Residual Biofilm after Decontamination and Biofilm Persistence

Following biofilm formation, the disks were split into 3 groups, and the decontamination was performed. Controls (untreated) and treated disks (exposed to glycine or tagatose for 30 s/surface, at 5 mm distance) were transferred into new wells containing fresh medium and immediately assessed for post-treatment residual biofilm. Then, the plate was further incubated at 35 °C. At time 30 h, the disks were washed with PBS, transferred into new wells and assessed for residual bioluminescence by the Fluoroskan reader. The recorded luminescence signal was taken as measure of biofilm persistence.

### 4.2. Human DPSCs Isolation and Immune Selection

The study was carried out in accordance with the recommendations of the ethics committee of the province of Modena (Italy) (ref. number 3299/CE, September 5th 2017). Human DPSCs were isolated from third molars of adult subjects (*n* = 3; 30 to 35 years old) during routine extraction procedures after obtaining their written informed consent in compliance with the Declaration of Helsinki. Cells were isolated from dental pulp as previously described by Zordani et al. [30]. To briefly describe the procedure, after the extraction of the dental pulp, it was harvested from the teeth and underwent enzymatic digestion by using a digestive solution, (3 mg/mL type I collagenase plus 4 mg/mL dispase in α-MEM). A cell suspension was first obtained by filtering pulp onto 100 µm Falcon Cell Strainers, plated in 25 cm^2^ culture flasks and then expanded in standard culture medium (α-MEM supplemented with 10% heat-inactivated fetal bovine serum (FBS), 2 mM l-glutamine, 100 U/mL penicillin, 100 µg/mL streptomycin) at 37 °C and 5% CO_2_. An immune-selection was carried out on hDPSCs previously expanded by using MACS^®^ separation kit according to manufacturers’ instructions. Mouse IgM anti-STRO-1 and rabbit IgG anti-c-Kit (Santa Cruz, Dallas, TX, USA) were used as primary antibodies and then revealed by magnetically labeled secondary antibodies: anti-mouse IgM and anti-rabbit IgG (Miltenyi Biotec, Bergisch Gladbach, Germany). The immune-selection allowed the isolation of a homogeneous hDPSCs population expressing STRO-1 and c-Kit. All the experiments were performed using STRO-1+/c-Kit+ hDPSCs.

### 4.3. Human DPSCs Characterization

Immune-phenotypical characterization was performed by FACS analysis on immune-selected hDPSCs at passage 1. The expression of the typical mesenchymal stem cells (MSCs) markers, i.e., CD73, CD90, CD105, CD34, CD45, HLA-DR, was evaluated, as previously described by Pisciotta et al. [27]. Cells were stained with the following fluorochrome-conjugated antibodies (Abs): anti-human-CD73-PE-CY7, -CD90-FITC, -CD105-APC, -CD45-PE and -HLADR-PE-CY7 (all from BD Biosciences, Franklin Lakes, NJ, USA), and -CD34-ECD (Beckman Coulter, Fullerton, CA, USA). A minimum of 10,000 cells per sample was acquired and analyzed by using the Attune Acoustic Focusing Flow Cytometer (Attune NxT, Thermo Fisher, Waltham, MA, USA). Data was analyzed by FlowJo 9.5.7 (Treestar, Inc., Ashland, OR, USA) under MacOS 10.

### 4.4. Titanium Surfaces Characterization

Sandblasted titanium disks (Mectron spa, Carasco, Genova, Italy) measuring 6 mm in diameter and 3 mm in thickness were used in this study. In order to perform the experiments titanium disks, previously air-polished with two different experimental powders were divided into 3 experimental groups: (1) control group consisting of untreated surfaces (CTRL), (2) glycine group (GLY), (3) tagatose group (TAGAT). The cleansing procedure of all treated disks was carried out using the Air-Polishing System (Combi-Touch, Mectron spa, Carasco, Genova, Italy) with the two different powders. Briefly, the “Combi-Touch” air polishing system with the respective cleansing powder was used for 30 s at a distance of 5 mm according to the manufacturer instructions. In particular, the operating principle of “Combi Touch” air-polishing system consists in the mechanical action of compressed air spreading an accelerated flow of particles onto the titanium surface. When hitting the surface, the particles dissipate their kinetic energy almost completely, thus producing a gentle but effectively cleansing action. The cleaning treatment is completed by a water jet that is arranged in the form of a bell around the main flow and that uses the pressure drop originated around the nozzle to prevent the powder cloud from bouncing and being dispelled and, at the same time, to dissolve the powder by washing the surface [19]. After the air-polishing treatments, scanning electron microscopy analysis (EVO MA 10-Carl Zeiss, Oberkochen, Germany) was carried out to evaluate from a qualitative point of view the surface nanotopography for each experimental group.

### 4.5. Cell Morphology and Proliferation

Undifferentiated STRO-1^+^/c-Kit^+^ hDPSCs were seeded at a density of 1.5 × 10^3^ cell/cm^2^ on titanium disks in 48-multiwell units and expanded under standard culture conditions. After 24, 48, 72 h and 7 days of culture, cells were fixed in ice-cold paraformaldehyde 4% for 15 min without dissociating them from the titanium disks. The cells were subsequently permeabilized with 0.1% Triton X-100 in PBS for 5 min, stained with AlexaFluor546 Phalloidin (Abcam, Cambridge, UK) and rinsed with PBS 1%. Nuclei were stained with 1 µg/mL 40,6-diamidino-2-phenylindole (DAPI) in PBS 1%. Titanium disks were mounted with DABCO (Sigma-Aldrich, St. Louis, MO, USA) anti-fading medium on cover glasses. Cell proliferation and morphology were investigated using confocal microscopy (Nikon A1, Nikon, Tokyo, Japan), as formerly described by Di Tinco et al. [56]. Cell proliferation was measured by counting the DAPI-positive nuclei on 5 randomly selected fields measuring 4.4 × 10^4^ µm^2^ on 3 disks for each experimental group by a blind operator.

### 4.6. Evaluation of Stemness Markers in hDPSCs Cultured on Titanium Disks

After seeding hDPSCs at a density of 1.5 × 10^3^ cell/cm^2^ on titanium disks in 48-multiwell units and after 48 h of culture on each disk, cells were fixed in 4% ice-cold paraformaldehyde in PBS for 15 min and then processed as described above. The presence of the stemness markers STRO-1 and c-Kit was evaluated by immunofluorescence analysis in which the following primary Abs diluted 1:100 were used: mouse IgM anti-STRO-1 and rabbit IgG anti-c-Kit (Santa Cruz, Dallas, TX, USA). Secondary Abs (goat anti-mouse IgM Alexa488, goat anti-rabbit Alexa546) were diluted 1:200 (Thermo Fisher Scientific, Waltham, MA, USA). Nuclei were stained with 1 µg/mL 40,6-diamidino-2-phenylindole (DAPI) in PBS 1%. The multi-labeling immunofluorescence experiments were carried out avoiding cross-reactions between primary and secondary Abs. Confocal imaging was performed using a Nikon A1 confocal laser scanning microscope, as previously described. The confocal serial sections were processed with ImageJ software (NIH, Bethesda, MD, USA) in order to obtain 3-dimensional projections and image rendering was performed by Adobe Photoshop Software.

### 4.7. Analysis of VEGF Expression in hDPSCs Seeded on Titanium Disks after Different Air-Polishing Treatments

The expression of VEGF was investigated in hDPSCs seeded on titanium disks, following air-polishing treatment with different cleansing powders. Confocal immunofluorescence analysis of hDPSCs stained by mouse anti-VEGF antibody (1:100; Invitrogen, Carlsbad, CA, USA) was performed as described above.

### 4.8. Osteogenic Commitment

For the purpose of evaluating how osteogenic differentiation could be influenced by titanium surfaces, cells were seeded at 2 × 10^4^ cells/cm^2^ on these disks. After 72 h of culture, the standard culture medium was replaced with the osteogenic medium (α-MEM, 10% FBS, 2 mM L-glutamine, 100 U/mL penicillin, 100 mg/mL di streptomycin, 100 nM dexamethasone, 10 mM di β-glycerophosphate, all from Sigma-Aldrich, St. Louis, MO, USA). After 3 weeks of induction, immunofluorescence analysis was performed in order to assess the expression of typical differentiation markers, such as RUNX-2 and osteocalcin (OCN). The following primary antibodies were used, at a 1:50 dilution: rabbit anti-RUNX-2, mouse anti-OCN (Abcam, Cambridge, UK).

### 4.9. Statistical Analysis

All the experiments were performed in triplicate. Data were expressed as mean ± Standard Error (SEM). Differences among groups were analyzed by ANOVA followed by Newman-Keuls (cell proliferation) and Dunnett (residual biofilm evaluation) post hoc tests (GraphPad Prism Software version 5 Inc., San Diego, CA, USA). In any case, *p*-value < 0.05 was considered statistically significant.

## Figures and Tables

**Figure 1 ijms-22-00865-f001:**
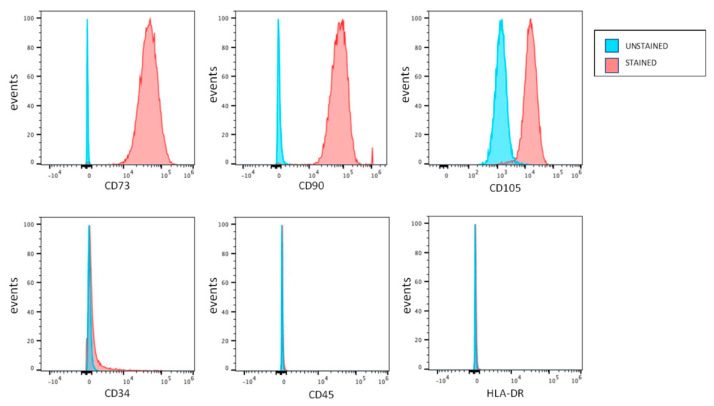
Immunophenotypic characterization of hDPSCs. Immunophenotypic characterization of c-Kit^+^/STRO-1^+^ hDPSCs by FACS analysis against mesenchymal surface antigens.

**Figure 2 ijms-22-00865-f002:**
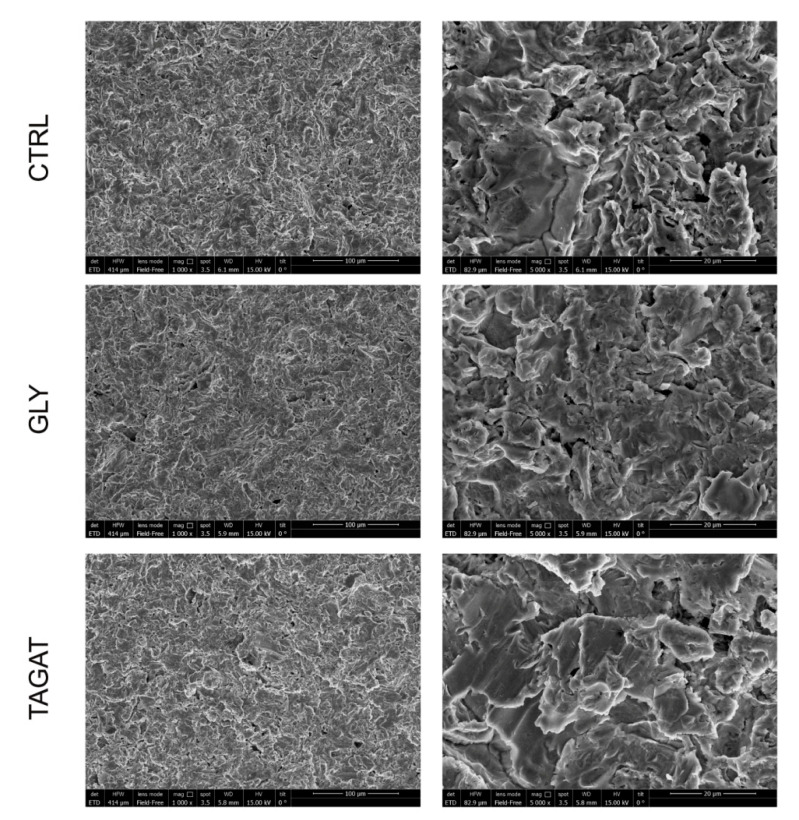
Titanium surface topography. Scanning electron microscopy (SEM) analysis at different magnifications was carried out on titanium surfaces from the three experimental groups (CTRL, GLY, TAGAT) in order to evaluate the surface topography. Scale bars: 100 µm (left) and 20 µm (magnifications on the right).

**Figure 3 ijms-22-00865-f003:**
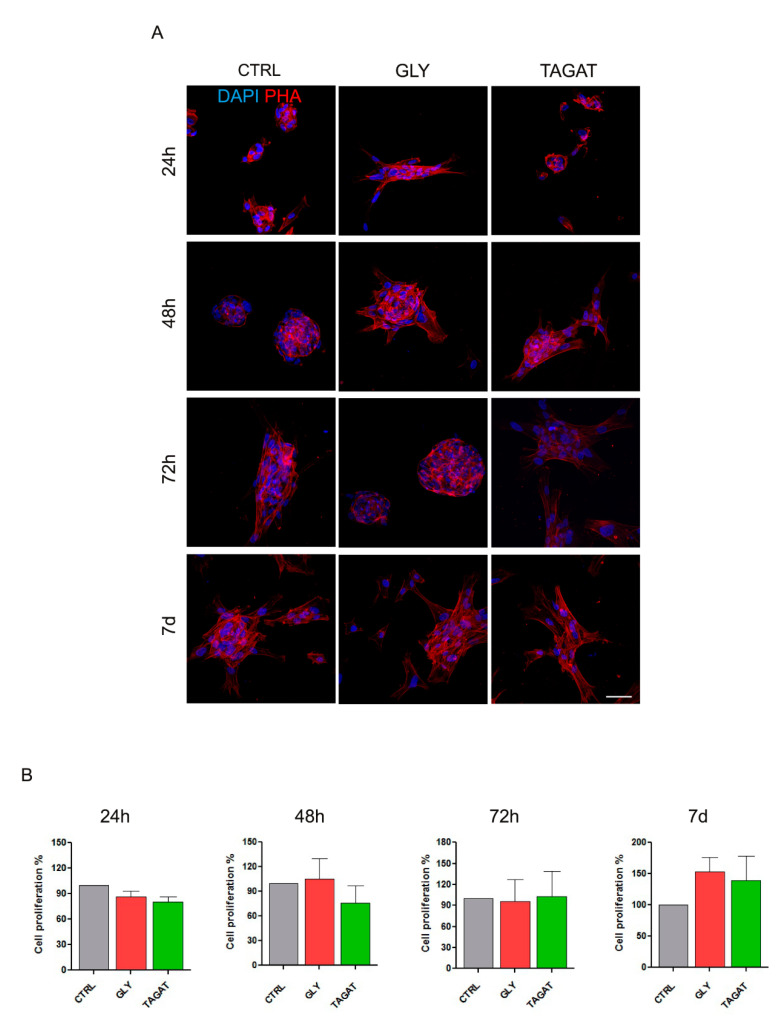
Stem cell morphology, adhesion and proliferation. (**A**) hDPSCs morphology was evaluated by immunofluorescence analysis of phalloidin (PHA)-stained cells at different time points (24 h, 48 h, 72 h, 7 d) on titanium surfaces previously cleansed with GLY and TAGAT. Control group (CTRL) consisting of hDPSCs seeded on untreated titanium surfaces. Nuclei were counterstained with DAPI. Scale bar = 50 µm. (**B**) Histograms represent cell proliferation (24 h, 48 h, 72 h, 7 d) of hDPSCs seeded on disks from different experimental groups and one-way ANOVA followed by Newman-Keuls post hoc test. Experiments were performed in triplicate.

**Figure 4 ijms-22-00865-f004:**
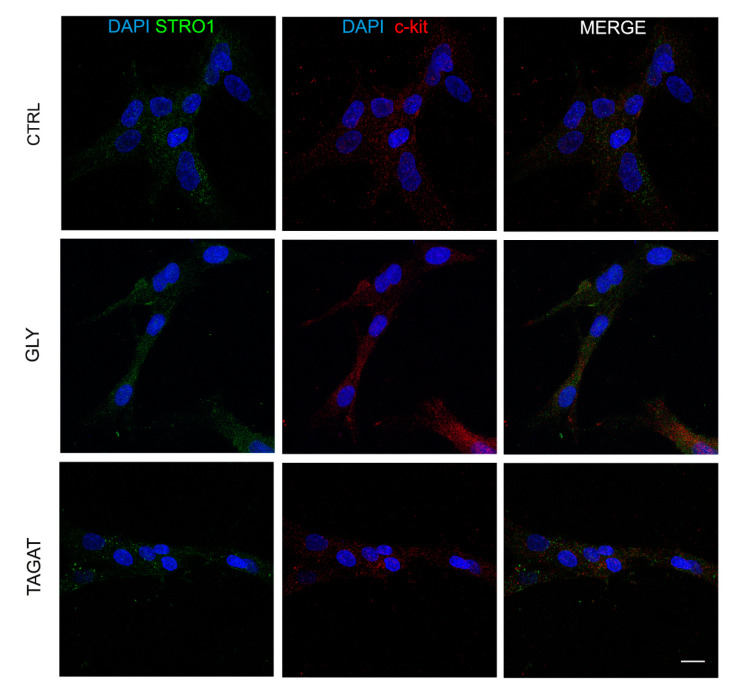
Assessment of stemness markers on immunoselected hDPSCs. Evaluation by immunofluorescence analysis of stemness markers STRO-1 (green) and c-Kit (red) after 48 h of culture on hDPSCs seeded on titanium disks previously cleansed with GLY and TAGAT. Control group (CTRL) consisting of hDPSCs seeded on untreated titanium surfaces. Nuclei were counterstained with DAPI. Scale bar = 10 µm.

**Figure 5 ijms-22-00865-f005:**
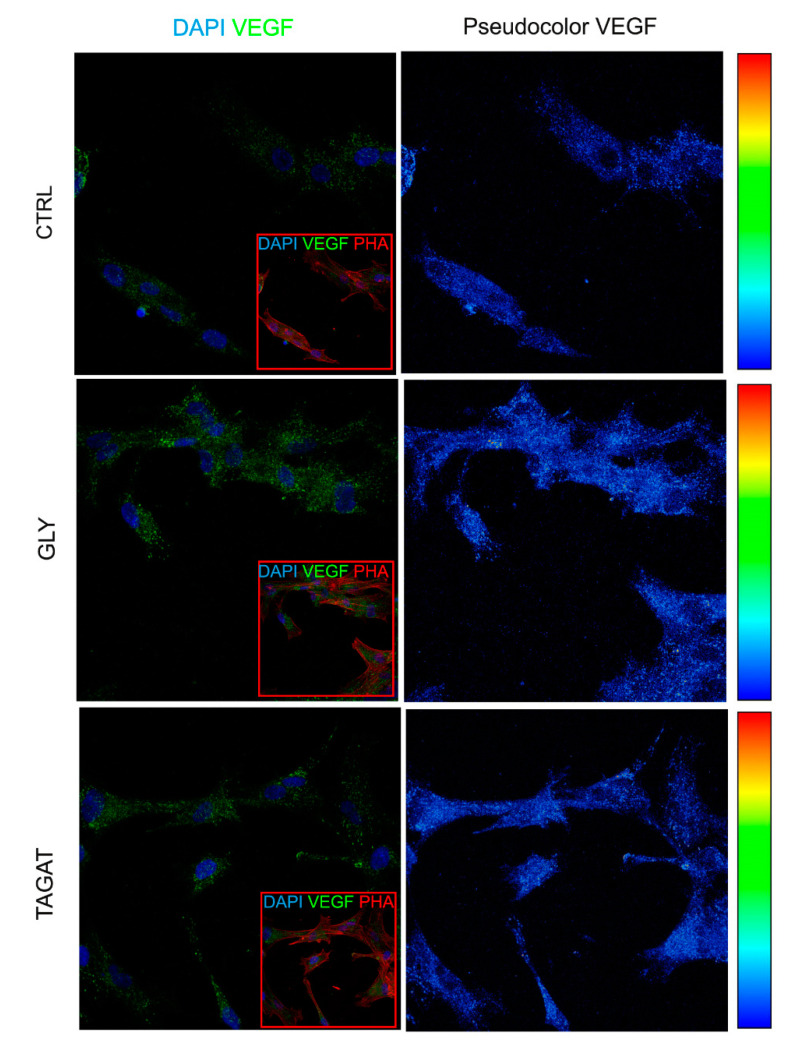
Angiogenic phenotype of hDPSCs after different air-polishing treatments. Analysis of VEGF expression by immunofluorescence analysis on hDPSCs seeded on titanium disks previously cleansed with GLY and TAGAT. Control group (CTRL) consisting of hDPSCs seeded on untreated titanium surfaces. Red squares show the morphology of hDPSCs after staining with phalloidin (PHA). Nuclei were counterstained with DAPI. Pseudocolor analysis of VEGF is shown on the right side. Scale bar = 10 µm.

**Figure 6 ijms-22-00865-f006:**
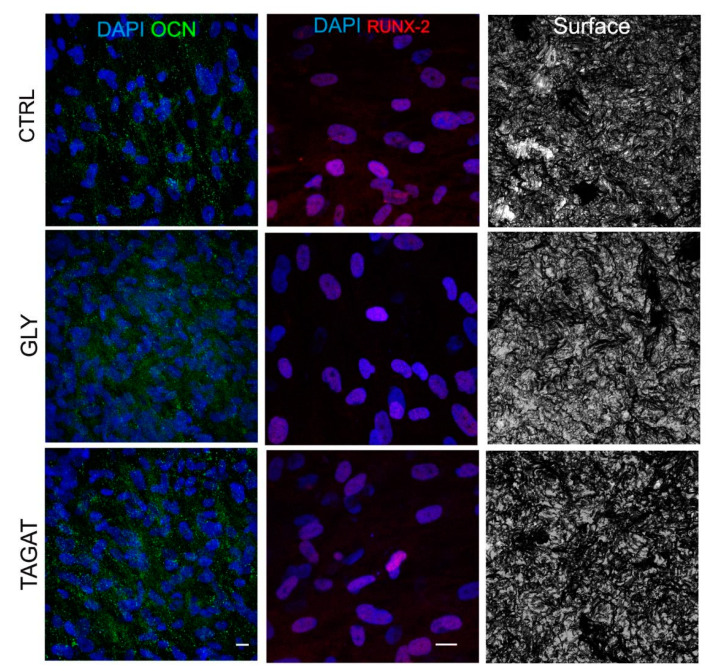
Osteogenic commitment of hDPSCs on titanium disks. Confocal immunofluorescence analysis of osteocalcin (OCN) and RUNX-2 were carried out on hDPSCs seeded on titanium disks previously cleansed with GLY and TAGAT. Control group (CTRL) consisting of hDPSCs seeded on untreated titanium surfaces. Nuclei were counterstained with DAPI. Scale bar = 10 µm.

**Table 1 ijms-22-00865-t001:** Evaluation of cleaning efficacy of the two polishing powders. Effects of the decontamination procedure on the formation and persistence of microbial biofilm onto titanium disks after air-polishing cleansing with glycine and tagatose. Control group (CTRL) consisting in titanium surfaces washed with saline solution. The values are express as RLU and represent the mean ± SEM of 5 replicates (* *p* < 0.05, GLY vs. CTRL and TAGAT vs. CTRL, when evaluating residual biofilm).

	RLU ± SEM			Fold Change		
Decontamination	Early Biofilm	Residual Biofilm	Persistent Biofilm	Residual Biofilm	Persistent Biofilm	Persistent Biofilm
Treatment	(pre-treatment)	(post treatment)	(30 h post treatment)	Treated vs. CTRL	Treated vs. CTRL	vs. Early biofilm
CTRL	0.0555 ± 0.010	0.1735 ± 0.059	1.7575 ± 0.548	-	-	+31.6
Glycine	0.0555 ± 0.040	0.0262 ± 0.001	1.0245 ± 0.467	−6.6	−1.7	+18.45
Tagatose	0.0555 ± 0.016	0.0429 ± 0.004	1.8504 ± 0.295	−4.0	−0.95	+33.34

## Data Availability

The data presented in this study are available upon request.

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
