# Peer review of "Evaluation of Antimicrobial Effect of Air-Polishing Treatments and Their Influence on Human Dental Pulp Stem Cells Seeded on Titanium Disks"

_ijms, 2021, doi:10.3390/ijms22020865_

Round 1

Reviewer 1 Report

The manuscript “Evaluation of antimicrobial effect of air-polishing treatments and their influence on human dental pulp stem cells seeded on titanium disks” proposes to investigate a novel experimental powder tailored for the development of subgingival prophylaxis treatments, in terms of effects on the biological properties of human dental pulp stem cells (hDPSCs). The authors found out that hDPSCs may provide a suitable cell source to assess the cell adhesion, when seeded nanostructured titanium disks, previously cleansed with different powders.

The work developed is clearly important from an academic and industrial point of view. It was well done, in general, with a wide range of characterization methodologies, including some statistical analysis. However, I have some questions / suggestions / changes to propose, not necessarily in this order: - In the topic of theoretical introduction, authors should dedicate a paragraph to explain the importance of topographic characterization methods, since the functionality of the Ti6Al-4V, surface depends strongly on the surface treatment, as they influence the anisotropy of the surface texture. The three-dimensional reconstruction techniques obtained through: 3D reconstruction using the extended depth of field method of optical microscopy, confocal laser scanning microscopy, or correlative microscopy*, are excellent approaches. I recommend that you add these informations, in the topic of the introduction, which is well explained in the cited article. - Add scales in Figure 1 B. - In the discussion topic, the authors state that according to the three-dimensional reconstruction shown in Figure 1B, there are no differences in the surface microtopography after the treatment with GLY or TAGAT, in relation to the control group. On what basis do you guarantee this? * Ana Horovistiz, Sílvia Carvalho, António J. Festas , J. Paulo Davim, Correlative microscopy analysis

of surface topography in machining Procedia CIRP 88 (2020) 565–569

Author Response

Reviewer 1

Open Review

(x) I would not like to sign my review report

( ) I would like to sign my review report

English language and style

( ) Extensive editing of English language and style required

( ) Moderate English changes required

(x) English language and style are fine/minor spell check required

( ) I don't feel qualified to judge about the English language and style

Yes         Can be improved            Must be improved         Not applicable

Does the introduction provide sufficient background and include all relevant references?

( )           (x)          ( )           ( )

Is the research design appropriate?

(x)          ( )           ( )           ( )

Are the methods adequately described?

(x)          ( )           ( )           ( )

Are the results clearly presented?

( )           (x)          ( )           ( )

Are the conclusions supported by the results?

(x)          ( )           ( )           ( )

Comments and Suggestions for Authors

The manuscript “Evaluation of antimicrobial effect of air-polishing treatments and their influence on human dental pulp stem cells seeded on titanium disks” proposes to investigate a novel experimental powder tailored for the development of subgingival prophylaxis treatments, in terms of effects on the biological properties of human dental pulp stem cells (hDPSCs). The authors found out that hDPSCs may provide a suitable cell source to assess the cell adhesion, when seeded nanostructured titanium disks, previously cleansed with different powders.

The work developed is clearly important from an academic and industrial point of view. It was well done, in general, with a wide range of characterization methodologies, including some statistical analysis. However, I have some questions / suggestions / changes to propose, not necessarily in this order: - In the topic of theoretical introduction, authors should dedicate a paragraph to explain the importance of topographic characterization methods, since the functionality of the Ti6Al-4V, surface depends strongly on the surface treatment, as they influence the anisotropy of the surface texture. The three-dimensional reconstruction techniques obtained through: 3D reconstruction using the extended depth of field method of optical microscopy, confocal laser scanning microscopy, or correlative microscopy*, are excellent approaches. I recommend that you add these informations, in the topic of the introduction, which is well explained in the cited article. - Add scales in Figure 1 B. - In the discussion topic, the authors state that according to the three-dimensional reconstruction shown in Figure 1B, there are no differences in the surface microtopography after the treatment with GLY or TAGAT, in relation to the control group. On what basis do you guarantee this? * Ana Horovistiz, Sílvia Carvalho, António J. Festas , J. Paulo Davim, Correlative microscopy analysis of surface topography in machining Procedia CIRP 88 (2020) 565–569

Re: We wish to thank the reviewer for the appreciation to our manuscript and for the constructive comments. As reported in our previous study (doi: 10.3390/ijms20081868) and highlighted by the interesting paper suggested by the reviewer, an optimal characterization of titanium surface and nanotopography can be reached  by combining different experimental techniques, i.e. AFM, SEM and confocal microscopy. To better support our data and conclusions we performed SEM analysis on titanium surfaces from each experimental group (CTRL, GLY and TAGAT). These data have been included in new Figure 2 in place of confocal microscopy 3D reconstruction images. The citation suggested by the reviewer was properly mentioned in our discussion.

Reviewer 2 Report

Why did the authors choose to evaluate biocompatibility in DPSC? This does not make sense, since we are approaching implants. In these situations these cells are never present. The authors justify it with a common embryonic origin, but why did they not use, for example, a pre-osteoblast cell line?

Page 11, line 30: the authors refer to “3 pós”, but only tested 2. Is this information correct?

Table 1, the authors should indicate with * ,for example, the statistically significant differences.

I suggested separating figure 1 into two figures. Since you are reporting different things, it makes no sense to be a panel.

Page 4, line 9: the abbreviations GLY and TAGAT appear for the first time here but they have been used before. Authors should refer the abbreviations previously.

Page 6, line 3: the authors want to refer figure 3 and not the 4, so they must correct it.

What is the interest in measuring VEGF expression? Again, it doesn't make sense in these cells because they are not addressing the dental pulp.

Statistical analysis: how was the normality / non-normality of the data assessed?

I suggested, in the results and in the materials and methods, to follow the same order of the experiences because it facilitates the comprehension of the readers.

For me, the first 3 paragraphs of the discussion should be moved to the introduction, since they relate to information already known and that justifies the work

Page 10, lines 34-36: the authors should remove this information, as this has not been evaluated.

Author Response

Reviewer 2

Open Review

(x) I would not like to sign my review report

( ) I would like to sign my review report

English language and style

( ) Extensive editing of English language and style required

(x) Moderate English changes required

( ) English language and style are fine/minor spell check required

( ) I don't feel qualified to judge about the English language and style

Yes         Can be improved            Must be improved         Not applicable

Does the introduction provide sufficient background and include all relevant references?

( )           (x)          ( )           ( )

Is the research design appropriate?

( )           (x)          ( )           ( )

Are the methods adequately described?

( )           (x)          ( )           ( )

Are the results clearly presented?

( )           (x)          ( )           ( )

Are the conclusions supported by the results?

( )           (x)          ( )           ( )

Comments and Suggestions for Authors

Why did the authors choose to evaluate biocompatibility in DPSC? This does not make sense, since we are approaching implants. In these situations these cells are never present. The authors justify it with a common embryonic origin, but why did they not use, for example, a pre-osteoblast cell line?

Re: We thank the reviewer for the interesting observation. As widely reported in literature, the successfulness of dental implants not only rely on osseointegration process but also on an effective vascularization of the implant itself. To this regard, it must be considered that human dental pulp stem cells, besides being derived from neural crest, are pericyte-like cells that are able to differentiate towards osteogenic lineage and, as demonstrated in vivo, to support an appreciable vascularization promotion in bone tissue regeneration (doi: Riccio et al. 2012 Tissue Eng, Pisciotta et al. 2012 Plos One). For these reasons, human dental pulp stem cells were chosen as a suitable cell source to study cells/implant interactions in vitro with particular focus on the effect of air-cleansing powders on cells biological properties. 

Page 11, line 30: the authors refer to “3 pós”, but only tested 2. Is this information correct?

Re: The oversight was corrected.

Table 1, the authors should indicate with * ,for example, the statistically significant differences.

Re: Statistically significant differences have been included in table 1 caption and reported in 2.1 paragraph.

I suggested separating figure 1 into two figures. Since you are reporting different things, it makes no sense to be a panel.

Re: Figure 1 was divided in 2 separate figures.

Page 4, line 9: the abbreviations GLY and TAGAT appear for the first time here but they have been used before. Authors should refer the abbreviations previously.

Re: The abbreviations have been specified at their first appearance.

 Page 6, line 3: the authors want to refer figure 3 and not the 4, so they must correct it.

Re: The oversight was corrected.

What is the interest in measuring VEGF expression? Again, it doesn't make sense in these cells because they are not addressing the dental pulp.

Re: As well reported in previous literature and confirmed by recent findings (doi:10.3390/ijms21093242, doi:10.3389/fcell.2020.00315), vascularization is a key process during osteogenesis and bone restoration. To this purpose, evaluating the expression of the pro-angiogenic factor VEGF can be predictive of the therapeutic growth of new blood vessels surrounding the implant material, in the first phases of osseointegration. Since hDPSCs are pericyte-like cells expressing VEGF and in light of the functional relationship between osteogenesis and angiogenesis, our study further aimed to evaluate whether and how the angiogenic properties of hDPSCs might be affected by the treatment of titanium implants with the two studied air-cleansing powders.

Statistical analysis: how was the normality / non-normality of the data assessed?

Re: The normality of the data was assessed and confirmed by D’Agostino-Pearson omnibus normality test (GraphPad Prism 7.0). 

I suggested, in the results and in the materials and methods, to follow the same order of the experiences because it facilitates the comprehension of the readers.

Re: Thanks for the suggestion. We have edited the order of related paragraphs between results and M&M as recommended.

For me, the first 3 paragraphs of the discussion should be moved to the introduction, since they relate to information already known and that justifies the work

Re: We thank the reviewer for the suggestion, however the information reported in the first three paragraphs are necessary to re-introduce the issue of the study and better discuss the obtained results. The Introduction section is mainly focused on the goal of our study. 

Page 10, lines 34-36: the authors should remove this information, as this has not been evaluated.

Re: We thank the reviewer for the observation. We have rephrased the sentence.

Round 2

Reviewer 1 Report

The work has been improved, but it could be better.

Reviewer 2 Report

The authors improved the scientific quality of the article after the corrections performed proposed by the reviewer. Thus, the article is ready for acceptance for publication.